# Baseline Quality of Life of Physical Function Is Highly Relevant for Overall Survival in Advanced Rectal Cancer

**DOI:** 10.3390/healthcare10010141

**Published:** 2022-01-12

**Authors:** Tim Fitz, Christopher Sörgel, Sandra Rutzner, Markus Hecht, Rainer Fietkau, Luitpold V. Distel

**Affiliations:** 1Department of Radiation Oncology, Universitätsklinikum Erlangen, Friedrich-Alexander-Universität Erlangen-Nürnberg, 91054 Erlangen, Germany; christopher.soergel@fau.de (C.S.); sandra.rutzner@t-online.de (S.R.); markus.hecht@uk-erlangen.de (M.H.); rainer.fietkau@uk-erlangen.de (R.F.); 2Comprehensive Cancer Center Erlangen-EMN (CCC ER-EMN), 91054 Erlangen, Germany

**Keywords:** rectal cancer, physical function, radiochemotherapy, quality of life, age

## Abstract

In advanced rectal cancer, neoadjuvant radiochemotherapy and total mesorectal excision lead to long overall survival. The quality of life (QOL) of the patients is clearly related to the prognosis. Our question was whether the prognosis can be represented with only one question or one score from the QOL questionnaires. 360 consecutively recruited patients diagnosed with advanced rectal cancer were questioned during radiochemotherapy and a follow-up of 8 years. The questionnaires QLQ-C30 and QLQ-CR38 were used; 10 functional and 17 symptom scores were calculated. The functional score “physical function” and the symptom scores “fatigue”, “nausea and vomiting”, “pain” and “appetite loss” were highly prognostic (*p* < 0.001) for overall survival. “Physical function” was highly prognostic at all time points up to 1 year after starting therapy (*p* ≤ 0.001). The baseline “physical function” score divided the cohort into a favorable group with an 8-year overall survival rate of 70.4% versus an unfavorable group with 47.5%. In the multivariable analysis, baseline “physical function”, age and distant metastases were independent predictors of overall survival. The score “physical function” is a powerful unrelated risk factor for overall survival in patients with rectal cancer. Future analyses should study whether increased “physical function” after diagnosis could improve survival.

## 1. Introduction

In recent years and decades, QOL has taken on an increasingly important role in the treatment of patients. QOL receives particular attention in palliative care, for example, when weighing quality-adjusted life years against maximum possible survival time extension. However, the field of QOL goes far beyond palliative care and is increasingly taken into account in the evaluation of therapy results or in diagnostics.

Despite its great importance, there is still no gold standard for a comprehensive assessment of QOL. This is because not all contributing parameters can always be determined and, on the other hand, the evaluation of QOL depends on the subjective assessment of the individual and cannot be measured externally. For this reason, various questionnaires have been established that categorize QOL in order to visualize it and allow comparisons. Due to the good clinical manageability, we decided to use this method.

With the combination of surgery and radiochemotherapy (RCT), very potent treatment options for rectal cancer (RC) are available today. Under the drastic therapy measures, there is temporarily a significant reduction in the patient’s QOL, which generally quickly returns to the initial level after completion of the therapy [1,2]. However, RC is one of the most common cancers in Germany and several thousand patients still die each year [3]. Accordingly, our idea was to investigate whether statements on the patient’s prognosis can be made by analyzing the QOL in order to further optimize the therapy in the next step. Of great interest for clinical applicability in this regard was to find a single score or question to simplify the analysis. 

For several years, the protective effect of physical activity on the development of RC [4,5] and the positive influence on the outcome [4,6] have been known. The latest meta-analysis from January 2021 on this topic reconfirmed the association between physical activity and RC, and further showed positive influences of physical activity on 12 other tumor entities [7]. Limitations of the meta-analyses of recent years were repeatedly the comparability of the physical activity level [7,8,9]. Of particular interest was therefore the score “physical function”, which is calculated from 5 items from the questionnaires and does not require the assessment of physical exercise in minutes per week or metabolic equivalent of task (MET) in hours per week as physical activity does.

## 2. Patients and Methods

### 2.1. Patient Cohort

From May 2010 until June 2020, 360 consecutive RC patients who agreed to participate were included in a prospective longitudinal study. The inclusion criteria were advanced rectal cancer and neoadjuvant radiochemotherapy. Almost exclusively neoadjuvant treatment concepts are performed at our institution; therefore, only neoadjuvantly treated patients were included. The intention was to analyze the QOL and to study the prognostic value for overall survival. All individuals were recruited at the department for radiotherapy of the University Hospital Erlangen.

### 2.2. Patient-Reported Outcome

The QLQ-C30 and QLQ-CR38 questionnaires of the European Organization for Research Treatment of Cancer (EORTC) were used in the surveys. Links to the questionnaires and the evaluation manual are provided in the Appendix A. The QLQ-C30 questionnaire consists of 30 Likert scale queries (items), each with 4–7 response options. Additionally, the QLQ-CR38 (38 items) is specifically designed for RC patients with 4–5 response options. The 68 items of the two questionnaires were cumulated into 27 percentage scales (0–100%). These were split into symptom and functional scores. While the functional scores consist at least of 2 items, the symptom scores can be built out of several items or only represent one item. For the functional scales, a higher score means the patient is doing well in this category. In contrast, a higher score in the symptom category implies more complaints for the patient. All patients received these two questionnaires the day prior to beginning their RCT (day -1), 2 weeks later during the RCT, at the end of the RCT (week 5), in week 10 shortly prior to surgery and from then on in yearly follow-ups (Figure 1D). At the beginning of the therapy, the patients who fulfilled the inclusion criteria were informed about the study in a personal conversation and gave their written consent. The Ethics Review Committee of the University Hospital Erlangen approved the study including the use of patients’ individual data. Patients could fill in the questionnaires at one of their aftercare appointment or the questionnaires were sent by mail. The questionnaires were digitalized by typing them into Excel spreadsheets. The clinical characteristics were derived from the local tumor documentation system and medical records.

### 2.3. Statistical Analyses

All statistical analyses were performed using IBM SPSS version 24.0 (IBM Inc., Chicago, IL, USA). Optimal cutoff points for overall survival and clinicopathological characteristics were calculated by receiver operating characteristic (ROC) curve analysis. Overall survival was visualized by Kaplan-Meier plots and significance was assessed by the log-rank test. All survival data were censored after 8 years. Differences were tested by the *t*-tests and Levene’s tests. Cox regressions models were used to calculate hazard ratios of quality of life scores and clinicopathological characteristics. Covariates with *p* < 0.2 in univariate analysis were included in multivariate analyses. The proportional hazards assumption was verified by visual inspection of the log-minus-log curves. The *p*-values < 0.05 were considered to be statistically significant.

## 3. Results

### 3.1. Clinical Characteristics

The cohort contains 360 consecutively recruited rectal cancer (RC) patients. More than 2/3 were male, the remaining were female (29.2%). The average age at diagnosis was 62.5 years, with the youngest individual at age 15 and the oldest at age 86. The majority of the cohort had advanced tumor stages with 60.8% of T3 and 24.7% of T4. Of note, 49.4% had N1 and 21.7% N2 lymph nodes metastases and 36.7% of the patients had distant metastasis. Tumors were graded as G2 (76.1%) histological grading, while 5.3% were G1 and 15.8% G3. Ten patients had an unknown histological grading and for one individual the T and N-status was missing.

Of the patients, 95.6% received the full neoadjuvant radiation dose of 50.4 Gy with a standard dose of 1.8 Gy per fraction. In addition, the following guideline conforms chemotherapy regimens were used: 5-flurouracil/oxaliplatin (75.3%), FOLFIRI, FOLFOXIRI, XELOX and 5-fluorouracil. Four patients did not receive any chemotherapy. Additionally, 120 patients received regional deep hyperthermia. In 331 of the 360 cases, a total mesorectal resection was performed 40 days after the end of the radiochemotherapy (Table 1). The median follow-up and median survival of the cohort was 48 months, respectively. After 5 years, the overall survival was 74.0%, the recurrence-free 63.7% and the metastasis-free survival rates 60.8% (Figure 1A). In general, older (*p* = 0.001) and metastasis-positive (*p* < 0.001) patients had a shorter overall survival (Figure 1B,C).

### 3.2. Function and Symptom Scores

Patients were asked to complete questionnaires prior to the beginning of RCT (-1 d), at the end of the 2nd week of therapy (2 w) and at the end of RCT (5 w). The next time of questioning was before total mesorectal surgery (10 w), followed by annual questionnaires (1–8 y) (Figure 1D). Higher functional scores reflect a higher degree of functionality. In 8 of the 10 baseline functional scores, 50% of the patients reached a score of 60% or more and in 7 of the 10 scores 25% of the patients reached a score of 100%, thus not indicating any limitations. The exceptions are “emotional functioning” (median = 66.7%), “global health status” (58.3%) and “future perspective” (33.3%) (Figure 2A). Out of the 17 symptom scores, 14 have a median symptom burden of 25% or less. Only for “fatigue” (33.3%), “insomnia” (33.3%) and “diarrhea” (33.3%) patients indicated having more symptoms (Figure 2B).

Threshold values of all scores were evaluated by receiver operating characteristic (ROC) analysis. Function and symptom scores were analyzed by Kaplan-Meier using the log-rank test. The 5-year survival rate differences in these analyses are summarized in Figure 2C,D. The upper and the lower ends of the bars represent the difference in the 5-year survival rates of the Kaplan-Meier plots. Several functional scores along with “global health status”, “role function” and “physical function” have clear differences between the favorable and unfavorable group. The score “physical function” is most relevant of all functional scores. At day -1, the difference between the higher (5 y survival = 82.6%) and lower (5 y survival = 63%) scoring group is 19.6% percent points (*p* < 0.001) (Figure 2C). Various symptom scores also indicate that patients with more burdens have a lower 5-year survival rate. Especially, “fatigue”, “nausea”, “pain” and “appetite loss” were prognostically valuable (*p* < 0.001) (Figure 2D).

### 3.3. Physical Function Individual Questions

We selected “physical function” as the most outstanding functional score. The score is formed of 5 items, asking the patient if he has trouble doing strenuous activities, like carrying a heavy shopping bag or a suitcase, taking a long walk, taking a short walk outside of the house, need to stay in bed or a chair during the day and if help is required for eating, dressing, washing or using the toilet.

We were interested whether there are differences among the five items defining “physical function” or if one item has more impact on the final score and could possibly replace it. These 5 items were compared each at day -1 and week 10 (Figure 3A). Although differences seem to be very small in the boxplots and there is no difference between the median values, the mean values are slightly higher at day -1. Next, the independence of the individual questions was examined at the same dates. For both day -1 and week 10, variance homogeneity was determined with Levene’s test between the questions “strenuous activities” and “long walk” and “short walk” and “bed and chair time”. However, the evidence for variance homogeneity did not reach a significant result between the first two questions (day -1: *p* = 0.060 and week 10: *p* = 1.000). Only at day -1 the variance homogeneity between “short walk” and “bed and chair time” is significant (*p* = 0.024), at week 10 the significance cannot be confirmed (*p* = 0.095). All other questions are independent of one another (*p* ≤ 0.001). Additionally, the significance of the individual items for the prognosis of overall survival was studied. Figure 4 summarizes the Kaplan-Meier analyses for each question at day -1 and week 10. All analyses at all survey dates can be found in the Appendix A. Especially clear differences were found with the items “strenuous activities” and “long walk”. At day -1, the 8-year survival difference is 22.9% (*p* ≤ 0.001) for “strenuous activities” (Figure 4A) and 21% (*p* ≤ 0.001) for “long walk” (Figure 4B). In week 10, it is 22.5% (*p* ≤ 0.001) for “strenuous activities” (Figure 4F) and 28% (*p* ≤ 0.001) for “long walk” (Figure 4G).

### 3.4. Physical Function 

Next, the score “physical function” was studied over the time period of 5 years. The majority of patients reached a high score, with half of patients over 86.7% and three quarters over 65% at baseline prior to the beginning of RCT. During RCT, physical function continuously decreased from the day -1 to the 5th week (*p* = 0.001). At week 10, shortly prior to surgery, the score rises again distinctly (*p* = 0.001). Over the next 5 years, the “physical function” score tends to increase slowly without reaching the initial value (Figure 3B).

Furthermore, “physical function” was studied by use of ROC curves to discriminate scores of well versus poor “physical function” for Kaplan-Meier analysis at the different dates. When questioning prior to the beginning of the RCT at day -1, 70.4% of the patients with better “physical function” survived 8 years and only 47.5% in the unfavorable group, that results in a difference of 22.9% (*p* ≤ 0.001) (Figure 5A). The largest 8-year survival difference of 29.7% (*p* ≤ 0.001) was found shortly prior to surgery at week 10 (Figure 5D). The increased survival of the group with the higher “physical function” was evident at dates, but did not reach the significance level at week 5, the 2nd-year and 3rd-year follow-up (Figure 5A–I).

It was also studied whether a score within the top 10% further improves the prognosis. For this purpose, the advantageous group was divided again by examining patients with a score of 90% and more separately. There was no survival benefit compared to those with a score of 76.6–90% (*p* < 0.001) (Appendix A). The same should also be studied for the bottom 10%, but due to very few patients with such a low score, the range was extended to the bottom third. Within the first 4 years, the survival rate for patients with a score in the lower third decreases dramatically. After another 4 years, the survival rate of patients with a score of 33.3–76.6 continued to decrease, while the survival rate of patients with a score in the lower third stagnated. Therefore, the survival rate after 8 years is approximately the same (*p* < 0.001) in the two lower groups (Appendix A).

### 3.5. Change Score

Additionally, we were interested in whether the change scores would have a prognostic significance. Patients with deterioration of their “physical function” from day -1 to the 2nd week were assigned to one group and patients with improvement or no change were assigned to the second group. The same procedure was performed from day -1 against all subsequent dates and from the 10th week against all subsequent dates. These comparisons were analyzed with Kaplan-Meier and log-rank regarding the overall survival. None of these analyses achieved significant effects on patient survival (Appendix A).

### 3.6. Risk Factor Analysis

A Cox regression analysis was performed to determine how much influence each factor has on survival. Considered for the univariate analysis were sex, age, grading, TNM-status and “physical function”. The multivariable analysis was performed only with variables having a *p*-value of 0.2 or less. Only the variables M-status (HR = 4.67, 95% CI = 2.81–7.76, *p* < 0.001), age (HR = 2.05, 95% CI = 1.30–3.22, *p* = 0.002) and “physical function” (HR = 0.54, 95% CI = 0.35–0.85, *p* = 0.008) are independent risk factors for overall survival (Table 2).

Finally, the influence of a combination of physical function with metastasis or age was studied by Kaplan-Meier and log-rank. For this purpose, “physical function” at day -1 was combined with metastasis carriers, non-metastasis carriers, patients older than 66 years, patients at 66 years and younger (Figure 6A–D). The group with better “physical function” in each analysis has a favorable survival. There are clear differences among the metastasis-free (*p* < 0.001), young (*p* = 0.030) and old age (*p* = 0.001) groups, as well as a trend in the group of patients with metastases (*p* = 0.092). Young patients with good “physical function” have a far superior survival (8 y survival rate: 74%) compared to old patients with poor physical function (8 y: 36%) (*p* < 0.001) (Figure 6J). One step further, the most favorable combination, young age, metastasis-free and high “physical function”, was compared to the least favorable combination, old age, metastasis-positive and low “physical function”, and all other patients were assigned as intermediates (Figure 6I). Between the most and the least favorable group, a huge 8-year survival difference of 70.3% (*p* < 0.001) was found.

All previous analyses were also conducted with the “physical function” data from week 10 (Figure 6E–H,K,L). Almost identical results were achieved. With one difference, the comparison between metastasis carriers and the level of “physical function” was significant (*p* = 0.044). Factors such as sex (*p* = 0.467), T-status (*p* = 0.200), N-status (*p* = 0.632) and grading (*p* = 0.289) were not included in the previous analyses, as no survival benefit was shown for any of them (Appendix A).

## 4. Discussion

The main focus of this study was to analyze the score “physical function” and assess its potential impact on overall survival of RC patients. Before starting RCT, our patients had the same level of “physical function”, with a mean of 78.5%, as a general age-adjusted population (mean = 78.9%) in Germany [10].

All data collected during the multiple query dates show a higher survival rate for patients with higher “physical function” compared to those with lower “physical function”. The most significant results were obtained before the start of the RCT (day -1) or after the completed RCT shortly before surgery (week 10), with a maximum survival rate difference of 32.3% (*p* < 0.001) after 7 years. Regarding the question of the most suitable time point of the query, no final decision can be made yet. Almost identical significant strong results were obtained both at day -1 and at week 10 after the start of the RCT. Another consideration, whether one of the 5 questions that form “physical function” would be more sufficient by itself, could be rejected.

For the comparison of two groups both on day -1 and after 10 weeks, the threshold value of 76.6% stood out. To use “physical function” clinically, one will probably prefer to divide patients into more than 2 prognosis groups. Our results indicated that a score well above 76.6% does not confer any further survival benefit (*p* < 0.001). For further gradations below 76.6%, results are not clear. Within the first 4 years, patients with scores in the lower third survive significantly shorter than patients with scores from 33.3 to 76.6% (*p* < 0.001). During the next 4 years, the survival rate stagnates, while the survival rate of the group in the middle third continuously decreases and even falls below that of the group in the lower third. However, the stagnation of the survival rate can be doubted. From the beginning, the group was very small with only 26 patients; after 4 years only 6 and after 8 years only 4 patients were observed. This could have led to a bias in the survival rate.

To ensure that “physical function” was not influenced by other variables such as age, TNM status, grading or sex, Cox regression analysis was performed. These confirmed “physical function” as an independent risk factor for the prognostic survival (HR = 0.54, 95% CI = 0.35–0.85, *p* = 0.008). The influence on patient survival is also evident when comparing younger and older patients in combination with high or low “physical function”. There is an 8-year survival difference of 20.6% (*p* = 0.001) between the two age groups (Figure 1B). In combination with “physical function”, the older group with high “physical function” achieves a higher survival rate than the younger group with poor “physical function” (*p* = 0.001) (Figure 6D). We went one step further and combined the independent risk factors of age, metastatic status and PF. The resulting 8-year survival rate difference of 70.3% (*p* < 0.001) is enormous (Figure 6I).

Numerous studies and meta-analyses have shown a positive correlation between physical activity and overall survival [4,5,6,7,8,11,12,13,14,15,16]. Thus, physical activity not only reduces the development of rectal cancer [4,7], but is also associated with prolonged survival after diagnosis [4,6]. Recurrent difficulties for the meta-analyses were the comparability of the level of physical activity and subsequent categorization into low, moderate, and high physical activity of the studies assessed [9]. The common determination of physical activity was by physical exercise in minutes per week or metabolic equivalent of task (MET) in hours per week. However, “physical function” is less related to sports or exercises, but rather reflects performance in daily life, is much easier to determine with only 5 simple questions, and achieves even greater survival prognosis differences in our cohort.

It was also studied whether individual improvements in “physical function” between the different dates of this query lead to an increased survival, but no significant effects could be detected. However, this study was not designed for such an analysis. Only score changes between different dates were examined without the patient receiving any training from our side. Nevertheless, this could be investigated in further studies, as Zaorsky et al. and Schumacher et al. have shown that exercise therapy can be safely utilized during radiotherapy of various carcinomas and leads to increased “physical function” [17,18,19]. Furthermore, a number of meta-analyses have already shown that an improvement in physical activity after diagnosis has a positive effect on survival [6,11,14]. This leads to the assumption that, with training, improved “physical function” could have a positive effect on survival as well. 

Other noticeable findings of the study included the most common symptoms before the start of RCT such as fatigue, insomnia, diarrhea and a particularly low score for “future perspectives”. Fatigue, diarrhea and insomnia are among the more common symptoms of RC [20,21,22]. However, a symptom combination of fatigue, insomnia and poor future perspectives should always be considered as depression. Again, the data of the study are not sufficient to conduct further investigations due to the study design. Nevertheless, depressive episodes in the context of a tumor disease should always be under observation [20]. No further data were collected on education, complementary therapy, alcohol, smoking, opioid use or others. There are many influencing factors that can affect quality of life. However, our results are so clear that these modifying factors have only limited influence.

Possible limitations of this study could be attributed to the questionnaire-based self-assessment of the patients. The questions designed by EORTC try to standardize the response as much as possible. Questioning the patient on having difficulties in performing different activities is a good tool. However, the individual expectation of themselves or their respective age group can vary. On the other hand, individual differences should be equalized by cohort size.

Another characteristic of our cohort is the predominantly advanced tumor stages. Compared to the German average of patients suffering from rectal cancer, the proportion of UICC stages III and IV predominates here. At the same time, the 5-year overall survival rate in our cohort of 74% is above the German average despite the predominantly advanced tumor stages. Thus, according to the German Cancer Registry in 2018, the 5-year survival rate of patients with rectal cancer was 75% in women and 68% in men. [23]. However, as this study only compared patients within this cohort, it can be assumed that the deviations from the average have no impact on the results of this study.

## 5. Conclusions

Taken together, this study was able to identify “physical function” as a powerful independent risk factor for overall survival of rectal cancer patients. To the best of our knowledge, we are the first to work with the score “physical function”. Both “physical function” and physical activity evaluate physical performance of CRC patients, but “physical function” is simpler to assess and offers enhanced scientific comparability with even greater survival prognosis differences. Further investigations are necessary to determine whether, by training, increased “physical function” could improve prognosis or even prevent deaths. In addition, the “physical function” should be integrated in clinical decision algorithms and might enhance individual treatment of rectal cancer patients.

## Figures and Tables

**Figure 1 healthcare-10-00141-f001:**
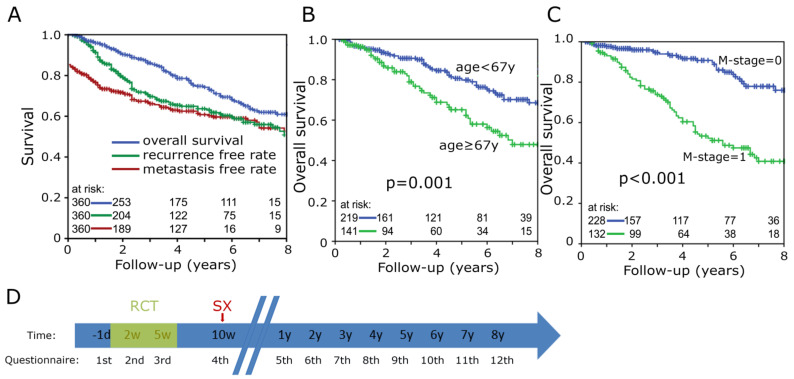
Kaplan-Meier plots for (**A**) Overall survival (upper blue line), recurrence-free rate (middle green line) and metastasis-free rate (bottom red line) of the entire cohort; (**B**) Cohort divided by age into below 67 years (upper blue line) and 67 years and above (bottom green line); (**C**) Metastasis status M = 0 (upper blue line) and M = 1 (bottom green line). (**D**) Timeline of each date when patients received questionnaire. Referring to the first day of radiochemotherapy (RCT), (SX) highlights day of surgery.

**Figure 2 healthcare-10-00141-f002:**
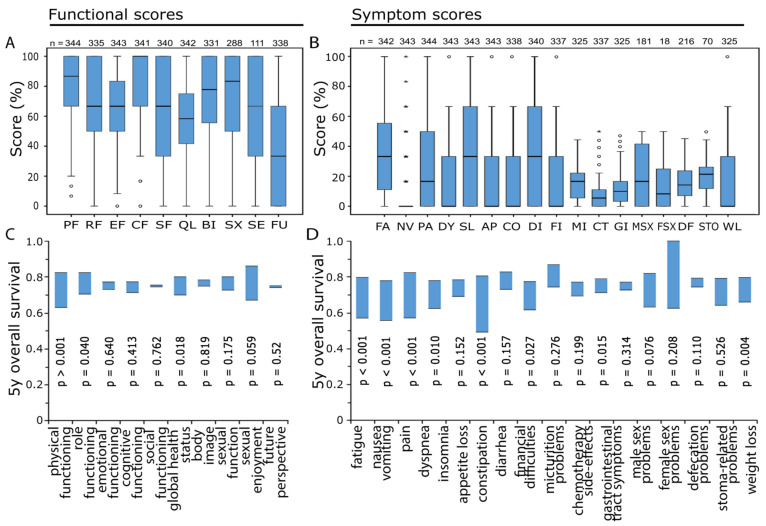
(**A**) Value distribution at day -1 for all functional scores, physical functioning (PF), role functioning (RF), emotional functioning (EF), cognitive functioning (CF), social functioning (SF), global health status (QL), body image (BI), sexual function (SX), sexual enjoyment (SE), future perspective (FU). (**B**) Result distribution at baseline for all symptom scores, fatigue (FA), nausea and vomiting (NV), pain (PA), dyspnea (DY), insomnia (SL), appetite loss (AP), constipation (CO), diarrhea (DI), financial difficulties (FI), micturition problems (MI), chemotherapy side effects (CT), gastrointestinal tract symptoms (GI), male sex problems (MSX), female sex problems (FSX), defecation problems (DF), stoma-related problems (STO), weight loss (WL). (**C**) Five-year survival rate in the Kaplan-Meier analysis of the groups with high and low functional scores. The upper end of the bar represents the 5-year overall survival of patients with high functional scores and the lower end that of patients with low functional scores. (**D**) Five-year survival rate in the Kaplan-Meier analysis of the groups with high and low symptom scores. The upper end of the bar represents the 5-year overall survival of patients with high symptom scores and the lower end that of patients with low symptom scores. The *p*-values indicate differences in overall survival by the Kaplan-Meier plots and log-rank test. “°” marks outliers and “*” marks extreme values.

**Figure 3 healthcare-10-00141-f003:**
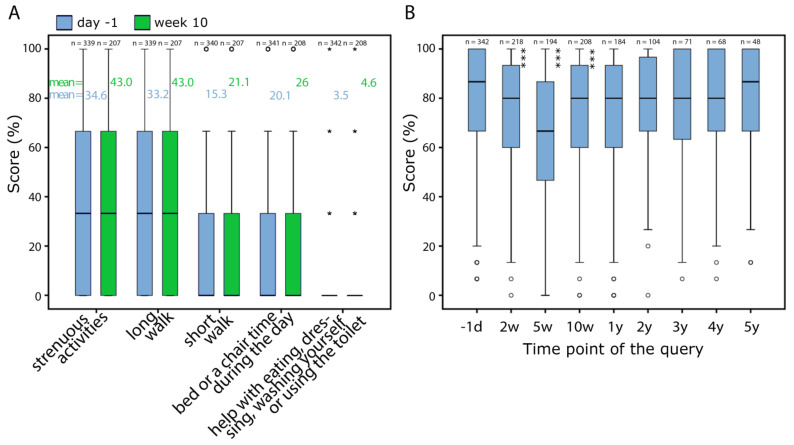
(**A**) Result distribution at day -1 (each left blue bar) and week 10 (each right green bar) for the 5 questionnaire items constructing “physical function”. (**B**) Result distribution of “physical function” at all dates. “°” marks outliers and “*” marks extreme values.

**Figure 4 healthcare-10-00141-f004:**
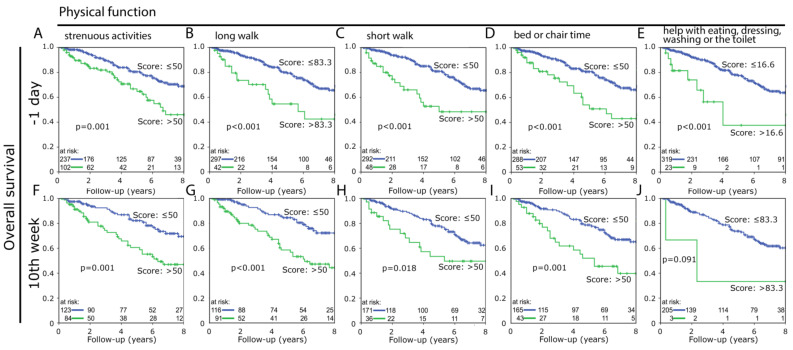
Overall survival between the group with lower (upper blue line) and higher (lower green line) scoring for the five questionnaire items representing “physical function”, with respective cutoff values at day -1 for (**A**) “strenuous activities”, (**B**) “long walk”, (**C**) “short walk”, (**D**) “bed or chair time”, (**E**) “help with eating, dressing, washing or the toilet” and week 10 for (**F**) “strenuous activities”, (**G**) “long walk”, (**H**) “short walk”, (**I**) “bed or chair time”, (**J**) “help with eating, dressing, washing or the toilet”.

**Figure 5 healthcare-10-00141-f005:**
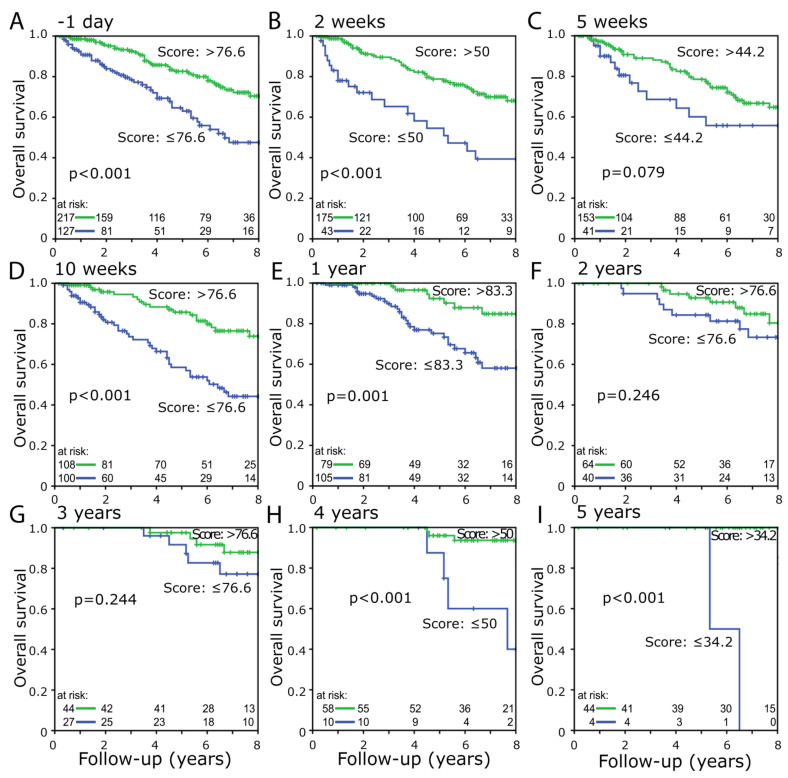
Overall survival between the group with higher (green line) and lower (blue line) scoring “physical function” with respective cutoff values at the different dates (**A**) day -1, (**B**) week 2, (**C**) week 5, (**D**) week 10, (**E**) 1 year, (**F**) 2 year, (**G**) 3 year, (**H**) 4 year, (**I**) 5 year.

**Figure 6 healthcare-10-00141-f006:**
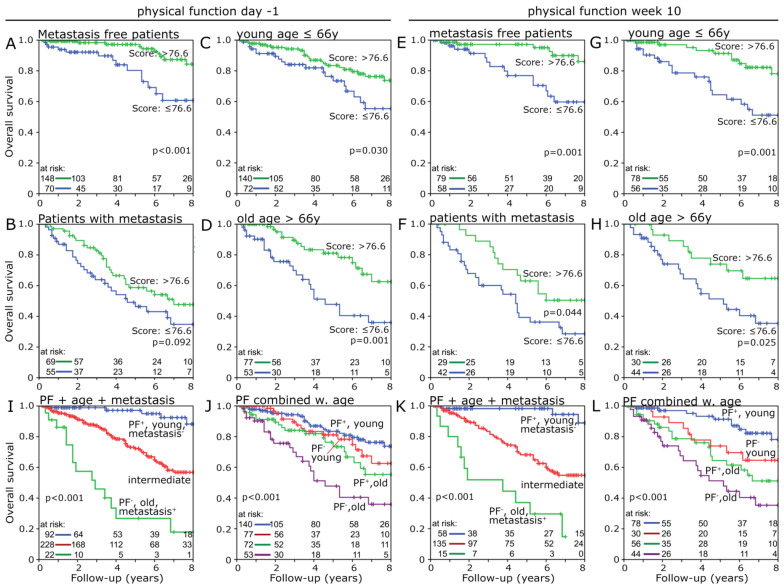
Overall survival at day -1 with higher (upper green line) and lower (bottom blue line) scoring “physical function” group in respective subgroups like (**A**) metastasis-free patients, (**B**) patients with metastasis, (**C**) young patients 66 years and below, (**D**) patients older than 66 years. Overall survival at week 10 with higher (upper green line) and lower (bottom blue line) scoring “physical function” group in respective subgroups like (**E**) metastasis-free patients, (**F**) patients with metastasis, (**G**) young patients 66 years and below, (**H**) patients older than 66 years. Overall survival at day -1 (**I**) and week 10 (**J**) comparing the combination (blue line) of high “physical function”, young age and metastasis-free (green line) with low “physical function”, high age and metastasis-positive and the intermediate group (red line), which include all patients with a mix of favorable and unfavorable attributes. Overall survival at day -1 (**K**) and week 10 (**L**) comparing all combinations of high and young age and high and low “physical function”.

**Table 1 healthcare-10-00141-t001:** Clinical characteristics.

Variable	Groups
Sex	female: 105 (29.2%); male: 255 (70.8%)
Age at diagnosis (yr)	mean: 62.47; min: 15; max: 86
T category	pT1: 11 (3.1%); pT2: 40 (11.1%); pT3: 219 (60.8%); pT4: 89 (24.7%)
N category	N0: 103 (28.6%); N1: 178 (49.4%); N2: 78 (21.7%)
M category	M0: 228 (63.3%); M1: 132 (36.7%)
Stage	UICC I: 24 (6.7%); UICC II: 46 (12.8%); UICC III: 157 (43.6%); UICC IV: 132 (36.7%)
Histological grading	G1: 19 (5.3%); G2: 274 (76.1%); G3: 57 (15.8%)
Radiation dose (Gy)	50.4: 344 (95.6%); >50.4: 16 (4.4%)
Chemotherapy	FOLFOX: 268 (74.4%); FOLFIRI: 18 (5%); FOLFOXIRI: 10 (2.8%); 5-FU solo: 52 (14.4%); XELOX: 8 (2.2%)
Surgery	Yes: 331 (91.9%); No: 29 (8.1%)

**Table 2 healthcare-10-00141-t002:** Univariate and multivariate analysis of overall survival according to Cox’s proportional hazards model.

	Univariate Analysis			Multivariate Analysis		
Variable	Hazard Ratio	95% C.I.	*p*	Hazard Ratio	95% C.I.	*p*
Physical function ≤ 76.6 [n = 119] vs. >76.6 [n = 213]	0.465	0.288–0.752	**0.002**	0.542	0.345–0.85	**0.008**
Sex f [n = 97] vs. m [n = 235]	1.639	0.927–2.896	0.089	1.486	0.867–2.547	0.150
Age <67 [n = 205] vs. ≥67 [n = 127]	1.931	1.219–3.06	**0.005**	2.047	1.302–3.216	**0.002**
T1/2/3 [n = 249] vs. T4 [n = 83]	1.092	0.641–1.862	0.746	---	---	---
N0 [n = 95] vs. N1/2 [n = 237]	0.768	0.469–1.257	0.293	---	---	---
M0 [n = 210] vs. M1 [n = 122]	4.566	2.727–7.645	**<0.001**	4.669	2.81–7.758	**<0.001**
G1/2 [n = 278] vs. G3 [n = 54]	1.572	0.886–2.789	0.122	1.538	0.876–2.699	0.134

## Data Availability

The datasets used and analyzed during the current study are available from the corresponding author on reasonable request.

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
