# Peer review of "Baseline Quality of Life of Physical Function Is Highly Relevant for Overall Survival in Advanced Rectal Cancer"

_healthcare, 2022, doi:10.3390/healthcare10010141_

Round 1

Author Response

Dear reviewer, thank you for the constructive criticism. We edited our manuscript according to your recommendations point by point. In the following, your comments are printed in italics and the insertions in the manuscript in red.

The authors present a very interesting work on the relevance of looking at the baseline quality of life of physical function of patients with advanced rectal cancer. They highlight the importance of this baseline quality of life of physical function on overall survival in advanced rectal cancer. This study is very well constructed and documented. The statistical analyses are sound and well conducted. The figures and supplementary figures are clear and easily understandable. Moreover, it is very original and relevant. This study shows the importance of taking care of the quality of life of patients. Just a remark: from an ethical point of view, do not write Materials and methods but Patients and Methods

We changed the headline from “Materials and Methods” in “Patients and Method”.

Reviewer 2 Report

In this article, the authors conducted the questionnaires QLQ-C30 and QLQ-CR38 with three hundred patients diagnosed with advanced rectal cancer during radiochemotherapy and a follow-up of 8 years in the German cohort. They concluded that baseline physical function, age, and distant metastases were independent predictors of overall survival in the multivariable analysis. It is a well-written paper. However, there are some concerns about this article.1.Similar articles have already been published previously—the authors should clearly state what is new in this article.2. The ratio of Stageâ…¢ and â…£ is high in this study population, which leads to a relatively high basis.3.Other variables, including education, alcohol consumptions, and opioid use, are considered clinical characteristics. 4. Precise explanations of QLQ-C30 and QLQ-CR38 are lacking (the authors may add the figures).

Author Response

Dear reviewer, thank you for the constructive criticism. We edited our manuscript according to your recommendations point by point. In the following, your comments are printed in italics and the insertions in the manuscript in red.

In this article, the authors conducted the questionnaires QLQ-C30 and QLQ-CR38 with three hundred patients diagnosed with advanced rectal cancer during radiochemotherapy and a follow-up of 8 years in the German cohort. They concluded that baseline physical function, age, and distant metastases were independent predictors of overall survival in the multivariable analysis. It is a well-written paper. However, there are some concerns about this article.

Please see below specific comments with regards to the submitted manuscript:

  1. Similar articles have already been published previously—the authors should clearly state what is new in this article.

Physical activity has only recently been identified as a prognostic factor. We assessed physical fitness differently than previously, but confirmed its utility as a prognostic factor. In the conclusion, the differences are highlighted by the following paragraph in the conclusion:

“To the best of our knowledge, we are the first to work with score “physical function". Both “physical function" and physical activity evaluate physical performance of CRC patients, but “physical function" is simpler to assess and offers enhanced scientific comparability with even greater survival prognosis differences.”

  1. The ratio of Stageâ…¢ and â…£ is high in this study population, which leads to a relatively high basis.

You are right, UICC stages III and IV predominate in our cohort. We recruited our patients at the department for radiotherapy and there were mainly patients in advanced tumor stages treated. At best, this should have a negative impact on the overall survival of the cohort. However, this does not affect the analyses within our cohort concerning physical function. We addend at the end of the discussion the following paragraph:

„ Another characteristic of our cohort is the predominantly advanced tumor stages. Compared to the German average of patients suffering from rectal cancer, the proportion of UICC stages III and IV predominates here. At the same time, the 5-year overall survival rate in our cohort of 74% is above the German average despite the predominantly advanced tumor stages. Thus, according to the German Cancer Registry in 2018, the 5-year survival rate of patients with rectal cancer was 75% in women and 68% in men. [23]. However, as this study only compared patients within this cohort, it can be assumed that the deviations from the average have no impact on the results of this study.”

  1. Other variables, including education, alcohol consumptions, and opioid use, are considered clinical characteristics.

You are right, that a lot of factors could influence QOL. Unfortunately, ten years ago, as we started the quality of life survey, we had not conducted such surveys and therefore cannot answer these questions.

We added in the discussion section at line 357:

“No further data were collected on education, complementary therapy, alcohol, smoking, opioid use or others. There are many influencing factors that can affect quality of life. However, our results are so clear that these modifying factors have only limited influence.”

  1. Precise explanations of QLQ-C30 and QLQ-CR38 are lacking (the authors may add the figures).

For copyright reasons, it is not possible to deposit the questionnaires in this manuscript. Therefore, we have included links directly to the English version of the QLQ-C30, QLQ-CR38, the page from which you can get the questionnaires in many different languages and to the English evaluation manual.

In the “Patients and Methods” section we stated:

“Links to the questionnaires and the evaluation manual are provided in the supplement.”

And in the Supplement we included:

“Supplementary “Patients and Methods”

Links to the different questionnaires and the manual:

Link to the English version of the QLQ-C30:

Link to the English version of the QLQ-CR38:

https://qol.eortc.org/?s=qlq-cr38

Link to the questionnaires and the user agreement

https://qol.eortc.org/questionnaires/

Link to the manual for the use and evaluation of the EORTC questionnaires

https://www.eortc.org/app/uploads/sites/2/2018/02/SCmanual.pdf”

The links were deposited on 30.12.2021.

Reviewer 3 Report

The manuscript "Baseline quality of life of physical function is highly relevant for overall survival in advanced rectal cancer" is a thorough article which helps highlight the importance of physical activity in advanced rectal cancer.
I have some minor comments for the authors:
- Was there a difference in the QOL in neoadj cases versus non-neoadj cases.
- Please explain your rationale for performing the study only on ne-adjuvant treated cancer cases

Author Response

Dear reviewer, thank you for the constructive criticism. We edited our manuscript according to your recommendations point by point. In the following, your comments are printed in italics and the insertions in the manuscript in red.

The manuscript "Baseline quality of life of physical function is highly relevant for overall survival in advanced rectal cancer" is a thorough article which helps highlight the importance of physical activity in advanced rectal cancer.

Please see below specific comments with regards to the submitted manuscript:

- Was there a difference in the QOL in neoadj cases versus non-neoadj cases.

- Please explain your rationale for performing the study only on ne-adjuvant treated cancer cases

Based on data from the CAO/ARO/AIO-04 trial, the Department of Radiation Oncology at Erlangen University Hospital decided to use only neoadjuvant therapies in the CAO/ARO/AIO-12 trial. We only studied patients from this center and therefore only data from neoadjuvant therapies were available. Therefore, we could not access adjuvant therapies at all.

In line 68 of “Patients and Methods” we state:

“The inclusion criteria were advanced rectal cancer and a neoadjuvant radiochemotherapy”

We added to the Patients and Methods section:

“Almost exclusively neoadjuvant treatment concepts are performed at our institution, therefore only neoadjuvantly treated patients were included.”

In line 118 of results we state:

“95.6% of the patients received the full neoadjuvant radiation dose of 50.4Gy”

Round 2

Reviewer 2 Report

The authors have responded to the reviewer's comments appropriately and well revised.